# My Family Accounts Much for Me: How Does Work-to-Family Conflict Lead to Unethical Pro-Family Behavior

**DOI:** 10.3390/ijerph20054368

**Published:** 2023-02-28

**Authors:** Yuming Wang, Wenan Hu, Zhaopeng Liu, Jinlian Luo

**Affiliations:** 1School of Economics and Management, Tongji University, Shanghai 200092, China; 2Shandong Institute of Talent Development Strategy, Shandong University, Jinan 250100, China; 3School of Management, Shandong University, Jinan 250100, China

**Keywords:** unethical pro-family behavior, work-to-family conflict, family motivation, self-determination theory, guilt proneness, ethical leadership

## Abstract

Although unethical behaviors are prevalent in the workplace, little is known about the unethical behavior aiming at benefiting one’s family (unethical pro-family behavior, UPFB). In this paper, we leverage self-determination theory to explore the association between work-to-family conflict and UPFB. Specifically, we hypothesize and corroborate a positive relationship between work-to-family conflict and UPFB, and such relationship is mediated by family motivation. Moreover, we identify two conditional factors, guilt proneness (at first stage) and ethical leadership (at second stage), in moderating the proposed relationship. In study 1 (scenario-based experiment, *N* = 118), we tested the causality of work-to-family conflict and intention to perform UPFB. In study 2 (field study, *N* = 255), we tested our hypotheses by employing a three-wave time-lagged survey design. As expected, results from two studies fully supported our predictions. Overall, we explain whether, how, and when work-to-family conflict will lead to UPFB. Implications of theory and practice are then discussed.

## 1. Introduction

Over the past decades, numerous corporate scandals and unethical incidents in the organizations emerged in an endless stream [1,2,3]. Management literature has corroborated the widespread existence of unethical behaviors in the workplace [4]. This is worrisome because unethical behaviors may cause damage to organizational reputations and thwart sustainable development of the enterprises [5]. Therefore, both scholars and practitioners have devoted substantial efforts to identify why unethical behaviors occur and seek optimal pathways to discipline them in the workplace [1].

Theoretically, there is a long-held understanding that unethical behaviors are driven primarily for pursuing self-interests [6,7]. Challenging this assumption, a burgeoning vein of research has elucidated that employees may occasionally perform unethical behaviors aiming at benefiting their families [1,3,5]. A novel construct capturing these behaviors has been labeled as unethical pro-family behavior (UPFB) [5]. UPFB refers to the actions which are conducted to benefit one’s family members whereas violating the societal or organizational moral norms [1,5]. Essential to this construct lies in its dual qualifications that it is immoral in nature on the one hand, but it is conducted intentionally to benefit actors’ families instead of themselves on the other hand [5]. Indeed, employees perpetrate a wide array of UPFB, such as taking organizational assets for family use, helping family members attain jobs in organizations irrespective of their family members are qualified or not, and so forth [5]. Although these types of misconducts are of prevalence in the workplace and cause huge losses to the organizations, scant research casts light into what propels employees to perform UPFB [5], discouraging the effort to inhibit it in the workplace.

Prior research has highlighted that unethical behaviors will be pronounced when employees experience role conflict [7]. Accordingly, under the current backdrop of fierce business competition, employees are particularly relevant to a specific form of role conflict, namely work-to-family conflict [8,9,10]. Work-to-family conflict captures the notion that role pressures from work domain thwart the fulfillment of family responsibilities [9,11]. Family responsibilities are fundamental considerations attach to human beings [12]. As work-to-family conflict prevents employees from accomplishing family responsibilities, we argue that employees failing to satisfy the basic needs psychologically (such as relatedness) [12] may potentially commit UPFB. However, due to the novelty of this construct, sparse research has empirically tested this proposition, constraining the scholarly understanding toward UPFB. 

In the purpose of advancing the emergent issue regarding UPFB, we invoke self-determination theory [13] to examine the potential relationship between work-to-family conflict and UPFB. Specifically, as self-determination theory highlights external inspirations, such as avoiding guilt or indebtedness, maintaining recognition when suffering from work-to-family conflict [11], will activate controlled motivation (e.g., family motivation), propelling employees to perform short-term focus behaviors [14,15], we theorize that involving work-to-family conflict will bear strong family motivation to expend effort to benefits family members, resulting in more UPFB.

Moreover, as employees’ family motivation is boosted to pursue specific extrinsic aspirations induced by work-to-family conflict, we anticipate that employees’ guilt proneness (i.e., the tendency to suffer from negativity and live up to normative expectations) [16], which indicates employees’ desire to avoid negativity and regain recognition when confronting work-to-family conflict, will intensify the effect of work-to-family conflict on family motivation. Finally, self-determination theory asserts that environmental factors will exert impacts on the process that motivations manifest its functions [15]. Drawing on the immoral nature of UPFB, we further argue that ethical leadership, which has salient moral implications in curtailing employees’ unethical conducts [17], can potently attenuate the relationship between family motivation and UPFB. 

Taking all these arguments into consideration, the current study employs self-determination theory to develop a two-staged moderated mediation model (Figure 1) and seeks to make the following contributions. First, by leveraging self-determination perspective, we unpack a new pathway for why UPFB will be conducted in the workplace. We echo previous contention that role conflict leads to unethical behavior and debunk that work-family conflict may result in UPFB through motivational pathway [7], broadening our knowledge about the potential motivators of UPFB in the workplace.

Second, invoking self-determination theory, we surmise that family motivation, which is more likely to be construed as a form of controlled motivation [12], will mediate the proposed relationship between work-to-family conflict and UPFB. By doing so, we not only shed important light on the potential intervening mechanism of why work-to-family conflict contributes to UPFB, but also respond scholars’ suggestions to advance the nomological network of family motivation [12,18]. Besides, expecting family motivation to positively predict UPFB also lends support to the potential downsides of family motivation [5,12]. 

Lastly, we expand the horizons of how distinct boundary factors influence individual’s motivational process. Based on family motivation literature [12,18] and self-determination theory [15], we identify two moderators, concerning both individual (i.e., guilt proneness) and contextual differences (ethical leadership), in shaping the psychological process from work-to-family conflict to UFPB. By doing so, we not only verify the contingency nature of motivational process [15], but also provide nuanced insight to strengthen or weaken these relationships at different stages within our theoretical model. Additional, by examining the conditional role of guilt proneness, we also advance a nascent stream of research that probes the short-term relational focus nature of guilt proneness, which challenges the consensus that guilt-related constructs may always lead to positive moral consequences [19].

## 2. Theory and Hypotheses

Self-determination theory articulates that the forms of work motivation experienced by individual can be constructed on a continuum ranging from autonomous to controlled [13]. To illustrated, autonomous motivation denotes behaving with senses of volition and possess the discretion of choice, whereas controlled motivation involves behaving with senses of pressure, senses of being obligated to participate in those activities [15]. More specifically, autonomous motivation encompasses motivations derive from enjoying the task itself (intrinsic motivation), valuing the object of the task (identified regulation), and synthesizing the task’s goal with the self [12,15]. Compared with autonomous motivation, motivations that stem from the desire to prevent guilt (introjected regulation) and approach external rewards or avoid punishments (external regulation) are generally perceived as more controlled [12]. 

Work motivations exert distinct impacts on employees’ attitudes, behaviors, and work consequences [20,21]. Autonomous motivation propels employees to be fully preoccupied with tasks with the senses of volition and enjoyment, which in turn lead to high levels of desirable work outcomes [13]. In contrast, controlled motivation constrains the scope of employees’ effort, potentially prompting employees to leverage short-term focused behaviors targeted desirable consequences [14]. Extant research employing self-determination theory mainly concentrates on examining autonomous motivation, such as how it can be enhanced through satisfying three basic needs and how it leads to constructive outcomes [20]. However, controlled motivation deserves more attentions from both academics and practice as it can also lead to desirable outcomes while generate unfavorable ones simultaneously [12,14]. 

Based on the lens of self-determination, family motivation, the desirable to expend effort to benefit one’s family [18], is prone to be deeded as relatively controlled [12]. Family motivation literature has demonstrated that family motivation tends to prompt employees to consider work as a means to afford and benefit their families [18], employees tend to feel controlled by such extrinsic aspirations [12]. Furthermore, expending efforts to benefit one’s family is more likely to perceived as employee “should” rather than he/she “want to” [12], involving a sense of be obligated to commit such actions instead of a sense of enjoyment to do so. Moreover, family motivation predisposes employees to be more concerned about the instrumental benefits that attach to their work rather than focusing the work itself [12]. Generally, this motivation is more inclined to be experienced as controlled. In the following parts, we will elaborate on how work-to-family conflict activates such controlled form of motivation (i.e., family motivation) and lead to short-term focused outcomes and boundary conditions.

### 2.1. The Direct Effect of Work-to-Family Conflict on UPFB

Work-to-family conflict describes situations that role demands from work and family domains are conflicting and employees choose work over family [9,10,22]. By definition, work-to-family conflict involves “devoting too much into work domains” whereas “devoting inadequately into family domains”. We argue that these two aspects can elicit different senses that directly contribute to UPFB.

First, work-to-family conflict is likely to predispose employees to feel indebted toward family members, which in turn contribute to UPFB. Specifically, the sense of indebtedness will be provoked when one party violated the reciprocity among two or more parties [23]. Stein [24] argued that accomplishing family-related responsibilities is generally regarded as societal expectations and norms. However, when involving work-to-family conflict, employees can’t satisfy family demands and fail to fulfill family-related obligations [11]. Such acts are considered as violating the inherent reciprocity among family members, which can elicit employees’ sense of indebtedness [25]. Furthermore, the sense of indebtedness is associated with short-term behaviors aiming at benefiting the indebted targets, as employees are urgently to get rid of such negative feelings [23]. As a consequence, work-to-family conflict will result in high levels of UPFB as it can serve as a shortcut for employees to satisfy the urgent needs (i.e., reconciling indebtedness as soon as possible). 

Second, work-to-family conflict tends to entitle employees at work domain and then facilitate them to perpetrate UPFB. Management research has claimed that employees will be entitled when they believe their devotions exceeds what they have received from the organizations [26]. In addition to failing to accomplish family-related obligations, work-to-family conflict also reflects that employees have devoted substantial resources into work domains as they choose work over family [10]. These devotions may entitle employees and enable them to consider that they deserve more superiority and privileges at work [27]. Further, the sense of entitlement tends to facilitate employees to engage in unethical acts because entitled employees truly believe they are owned and their actions are the way to get what they deserved [28]. Therefore, employees involving in work-to-family conflict may believe that they have the personal rights to benefit their families by conducting UPFB. 

In summary, work-to-family conflict is likely to increase employees’ senses of entitlement and indebtedness, resulting to more UPFB. Indeed, a wide array of research has asserted that work-to-family conflict is likely to trigger detrimental outcomes at work and increasing the probability for employees to perform unethical conducts [29,30]. Therefore, we hypothesize the following:

**Hypothesis** **1:**
*Work-to-family conflict is positively related to UPFB.*


### 2.2. Mediating Role of Family Motivation

According to self-determination theory [13,14], the types of motivation employees experience can be shaped by individuals’ aspirations. When employees strive to purse extrinsic aspirations, such as acquiring external rewards, avoiding guilt, retaining recognition, or preventing punishment, they are more likely to experience controlled forms of motivation. In contrast, autonomous types of motivation grow more dominant when employees are preoccupied with intrinsic aspirations, such as personal development and meaningful contributions [12]. Family motivation, as a specific form of controlled motivation, is likely to be activated when employees strive to achieve extrinsic aspirations pertained to family-related, such as preventing guilt toward families or maintaining family-related identity and recognition [12]. Accordingly, we propose that work-to-family conflict can trigger employees’ family-related extrinsic aspirations, which contribute to enhanced family motivation.

First, family motivation will be fueled for avoiding guilt and indebtedness toward employees’ families. The senses of guilt or indebtedness are likely to be generated when employees act in a way which is inconsistent with common attitudes, beliefs, or societal expectations [31]. Accordingly, when employees get stuck in work-to-family conflict, they can’t sufficiently accomplish the family-related responsibilities that are regarded as societal expectations and norms [22,24]. Thus, the probability for employees to induce guilt and indebtedness increases and employees may set a primary goal to avoid such negative emotions. Drawing on self-determination theory, when employees confront with an urgent needs to avoid the negativity, which indicates they are focusing on pursuing extrinsic aspiration at work, they are more likely to experience controlled forms of motivation [14]. Moreover, given that the targets employees feel guilty toward are their family members, employees’ family motivation are likely to be triggered to benefit their families as a way to satisfy their extrinsic aspirations.

Second, the inspirations to maintain the identity or recognition impaired by work-to-family conflict will also enhance employee’ family motivation. As work-to-family conflict prevents employees from fulfilling the family-related obligations they traditionally hold, employees may perceive they have violated the societal norms or “good family members” standards [11], which further results in concerns to their family-related identities and cause decreased recognition from others. However, employees have an inherent preference to maintain their identities and recognition from others in order to retain self-integrity [32]. Interfered by work-to-family conflict, employees may strive to acquire the extrinsic inspirations to maintain their family-related identities and reattain recognition. Thus, employees’ family motivation is likely to be boosted to benefit their families at work, as this can serve as a way for employees to maintain family-related identities and harvest the recognition as good family members [18].

Furthermore, a controlled form of motivation is suggested to narrow the range of employees’ effort, motivating employees to engage in specific behaviors, especially those can obtain short-term positive effects [14]. In particular, we contend the potential short-term behavior can be a specific unethical behavior aiming at benefiting family members (i.e., UPFB). Specifically, when family motivation is predominant, employees are more likely to portray that UPFB is in the service of a meaningful purpose [33]. In this vein, UPFB is more acceptable for employees who reconstruct it as a way for achieving meaningful goals. Wiltermuth [34] contended that when unethical behaviors aim at benefiting others, actors are more likely to justify their unethical conducts as pro-social or meaningful goals, lending support to our argument. Furthermore, benefiting one’s family is also considered as societal expectations, and employees who benefit their families is perceived as they “should” more than they “want” [18,24]. Hence, employees may conduct UPFB in the names of societal expectations, which can serve as ways to displace the responsibilities of UPFB [33]. Moreover, Menges et al. [18] has demonstrated that employees with strong family motivation give priority to their families. High family motivation predisposes employee to categorize family members over other groups (e.g., coworkers, shareholders, and stakeholders in the workplace), thereby cognitively valuing humanized persons whereas experiencing less moral self-sanction when hurting dehumanized ones [33], resulting in more UPFB.

Supporting our arguments, Nickerson [35] argued that employees seek to selectively notice, encode, and retain information that parallels their desires. When employees are driven by strong motivation to benefit their families, they tend to focus largely on the instrumental value of UPFB, resulting in more UPFB at work. Extant empirical research also lends support to our arguments. For instance, Chen et al. [6] has disclosed the pro-organization intention may lead to unethical pro-organizational behavior. Besides, an abundance of research has warned that strong family motivation may potentially lead to unethical behavior [5,18]. Collectively, we generate the following hypothesis:

**Hypothesis** **2:**
*Family motivation mediates the relationship between work-to-family conflict and UPFB.*


### 2.3. Moderating Role of Guilt Proneness

Guilt proneness denotes a tendency to suffer from negative feelings about personal misconduct, even when it is private [36]. As a relatively stable dispositional construct, guilt proneness captures two dimensions of focal individual’s guilty experience: intensity and frequency [37]. That is, employees high in guilt proneness are more prone (or have lower threshold) to experience guilt and tend to experience it more intensively when interfered by the same inducers. Extant literature has also demonstrated higher guilt proneness is strongly associated with more effort to avoid guilt and fulfill normative expectations [38]. In the current research, we propose that guilt proneness will strengthen the positive relationship between work-to-family conflict.

First, guilt proneness increases the extrinsic inspirations of avoiding those negative feelings triggered by work-to-family conflict, thereby driving higher family motivation. As illustrated above, family motivation will be boosted for employees strive to achieve the extrinsic aspirations triggered by work-to-family conflict, the more urgent the extrinsic aspiration it is, the higher the salience of family motivation will be [12]. Higher levels of guilt proneness are associated with higher probability to experience negative feelings resulted by work-to-family conflict [39]. In this regard, employees with high levels of guilt proneness devote more efforts to achieve their extrinsic aspirations than those employees low in guilt proneness. As higher levels of family-related extrinsic aspirations will contribute to enhanced family motivation, we contend that guilt proneness will accentuate the relationship between work-to-family conflict and family motivation. 

Second, guilt proneness can also enhance employees’ willingness to maintain the recognition concerning family issues. Employees choosing work over family will lead to concerns toward their family-related identity and decrease the recognition from others [10]. Prior research has highlighted guilt proneness is linked with higher convictions to live up to normative expectations to gain recognition [38]. Under such circumstance, high levels of guilt proneness promotes employees’ extrinsic inspirations to reconfirm family-related identity and retain recognition that impaired by work-to-family conflict. Thus, based on self-determination theory, such higher levels of extrinsic inspirations concerning family issues jointly shaped by work-to-family conflict and high guilt proneness will contribute to higher levels of family motivation than when possessing low guilt proneness.

In summary, we contend that guilt proneness will further boost employees’ extrinsic aspirations resulted by work-to-family conflict. That is, guilt proneness can promote the extrinsic aspirations of striving to prevent negative feelings and acquire the recognition as good family members, leading to enhanced family motivation. Taken together, the following hypotheses are put forward:

**Hypothesis** **3:**
*Guilt proneness moderates the relationship between work-to-family conflict and family motivation, such that the relationship is stronger when guilt proneness is high rather than when is low.*


**Hypothesis** **4:**
*Guilt proneness moderates the indirectly relationship between work-to-family conflict and UPFB via family motivation, such that the indirect relationship is stronger when guilt proneness is high rather than when is low.*


### 2.4. Moderating Role of Ethical Leadership

Self-determination theory asserts that environmental factors can exert consequential influences on the process where motivation manifests its impacts [13]. Ethical leadership, which captures the notion that supervisors put strong emphasis on normatively appropriate acts through person actions and interpersonal relationships, and seek to increase such conducts from subordinates through two-way communication, reinforcement, and decision-making [17], can serve as excellent ethical role model and form high ethical atmosphere to prohibit employees’ immoral acts [40]. Thus, we expect that ethical leadership can weaken the association between family motivation and UPFB.

First, ethical leaders serve as the ethical role model for employees to emulate, which are conducive to effectively regulate employees’ misconducts. Leaders occupy an important position in the organizations and thus easily to become role models [41]. Ethical leadership can demonstrate normatively proper conducts through personal actions [17], such as behaving in honest, fair, and moral manners in their personal and professional lives [42,43]. As they serve as role models for employees [41], ethical leadership can effectively improve employees’ ethics-related thoughts and behaviors, promoting the internalization of moral perspectives [17]. Therefore, employees will bear ethics in mind and devote more attention to the ethical nature of their behaviors when pursuing personal goals, which are conducive to prevent employees from conducting unethical and inappropriate behaviors [42]. As a consequence, although employees exhibiting high family motivation seeking to benefit their families, they are less likely to engage in misconducts under ethical leadership, which in turn attenuate the relationship between family motivation and UPFB.

Second, ethical leaders form highly ethical environment in the organization, which is beneficial for powerfully inhibiting morally unintended behaviors. Ethical leaders will draw heightened attention to ends as well as means [40]. In the aim to emphasize this value, ethical leaders will reward employees who achieve their goals by appropriate ways, whereas disciplining those obtaining goals via unethical conducts [41]. Besides, ethical leaders highlight the two-way communication between leaders and followers [17]. That is, they directly communicate with employees regarding organizational moral standards and behavioral expectations [43], which is conducive to establish ethical environment by fostering employees’ understanding of ethical norms [40]. When working with ethical leaders, employees explicitly understand the disciplines and are sensitive to the morality of their behaviors. Therefore, despite the exist of strong family motivation, employees will still concern about the morality of their behaviors, and thus devote more attention to judge the ethical nature of their behaviors. In this regard, when ethical leadership is high, the relationship between family motivation and UPFB will be weakened. Thus, we hypothesize the following:

**Hypothesis** **5:**
*Ethical leadership moderates the positive relationship between family motivation and UPFB, such relationship will be weaker when ethical leadership is high rather than when it is low.*


**Hypothesis** **6:**
*Ethical leadership moderates the indirect relationship between work-to-family conflict and UPFB via family motivation, such that the indirect relationship will be weaker when ethical leadership is high rather than when it is low.*


### 2.5. Studies Overview

Two empirical studies have been conducted to test our proposed hypotheses. In Study 1, the extent of work-to-family conflict has been manipulated to examine its effects on employees’ intention to perform UPFB, providing causality for the two. In Study 2, a three-wave and time-lagged field study has been conducted to examine our full conceptual model, including the mediating role of family motivation and the moderating roles of guilt proneness and ethical leadership. After integrating the results from different methodologies, the validity and generalizability of our research findings are more ensured. 

## 3. Study 1 Scenario Experiment

### 3.1. Participants and Procedure

120 participants who were full-time employed have been recruited through an online panel data platform, Credamo (https://credamo.com/ (accessed on 22 May 2022)), which is equivalent to Prolific. We first announced a consent to all these participants before starting the experiment, which articulated that the experiment was conducted with academic purposes and all the information provided would be kept seriously confidential, and participants could engage in it voluntarily to attain a small sum of compensation (2 yuan, i.e., 0.3 dollars). The average age of the participants was 30.96 years (SD = 4.52), ranging from 21 to 47. 47.46% of the participants were male. Most of the participants held a bachelor’s degree (79.66%). More detailed statistics were provided in Table 1. Statistical calculations were conducted in SPSS 26.0 (designed by IBM, New York, NY, USA). Consistent with most of prior research [5,12], the statistics (mean and deviation) of category variables (e.g., gender, education) aim at disclosing their distributions and with no real meanings, while the statistics of the continual variables (e.g., age or our focal variables) reflect the real value of means and deviations calculated from the accurate values or the scores from Likert scale. The correlational coefficients represent the statistical relevance between two tested variables, which can provide preliminary examinations about our hypotheses.

120 participants were randomly assigned to two scenarios which manipulated either high or low levels of work-to-family conflict. Before starting the scenario experiment, participants reported their demographic information. Further, they were instructed to carefully read the scenarios and immerse in these situations. 2 questionnaires were excluded because there were apparently careless responses (i.e., 2 year-old and 27 children at home), resulting in 118 valid questionnaires eventually (58 in high-level of work-to-family conflict condition and 60 in low-level of work-to-family conflict condition).

### 3.2. Scenario Instructions

Scenario instructions were designed via adopting the scenarios of work-to-family conflict developed by Greenhaus and Powell [44], which are considered as the most commonly used scenarios in work-to-family conflict literature [10]. The core difference between these two scenarios is that the importance of affairs interfered by work is varied. We presented the instructions in Chinese. In high-level of work-to-family conflict, the scenarios stated that, *it is Sunday today, and it is also the birthday of one of your family members. You have already prepared a ceremonial dinner party for all your family members to celebrate this important day. However, on Sunday morning, you received an emergency message from company that informed you to start a business trip to another city right now. Therefore, you had to miss the family dinner party.*

In regard to the low-level work-to-family conflict condition, the instruction stated that, it is a workday today. You have planned to go home for dinner with family members as usual. However, at the moment you were leaving for home, you were asked to work overtime to finish an emergency task. Therefore, you had to inform your family members that you could not have dinner with them tonight.

### 3.3. Measures

Considering the participants’ native language, we employed translation-back translation procedure proposed by Brislin [45] to ensure the scales’ semantic equivalence with original scales. Participants responded to all the items with 5-point Likert scale (1 = “strongly disagree” to 5 = “strongly agree”).

***Intention to**perform UPFB*.** Intention to perform UPFB was accessed by revising the 7-item scale developed by Liu et al. [5] to fit the current scenario contexts. Sample item includes “To help my family, I would take company asserts/supplies home for family use.”

***Control variables*.** As our research involved both work and family domains, we followed previous scholars advice to control participants’ age, gender (0 = female, 1 = male), education (1 = technical college or below, 2 = bachelor, 3 = master, 4 = doctor), organizational tenure (1 = tenure ≤ 0.5 years; 2 = 0.5 year < tenure ≤ 1 year; 3 = 1 year < tenure ≤ 4 years; 4 = 4 years < tenure ≤ 10 years; 5 = tenure > 10 years), marital status (1 = unmarried, no partner; 2 = unmarried, living with partner; 3 = married; 4 = divorced; 5 = widowed), and the number of children at home [5,46].

### 3.4. Manipulation Checks

In the aim to ensure the effectiveness of our manipulations, we asked the participants to rate, “To what extent do you perceive the situation described by the above scenario instruction a critical conflict to your family life?” with a 5-point Likert scale [10]. An independent-sample T-test in SPSS 26.0, which revealed that participants assigned to the high-level of work-to-family conflict condition (M = 4.26, SD = 0.66) perceived significantly higher degrees of work-to-family conflict than those assigned to low-level of work-to-family condition (M = 3.72, SD = 0.76; *t* = 4.12, *p* < 0.001), indicating successful manipulations.

### 3.5. Results of the Experiment

Another independent-sample T-test was conducted in SPSS 26.0 to statistically compare the mean scores of intention to perform UPFB between the two conditions, which demonstrated intention to perform UPFB was significantly higher in high-level of work-to-family conflict condition (M = 1.99, SD = 0.80) than in low-level of work-to-family conflict condition (M = 1.68, SD = 0.44; *t* = 2.60, *p* < 0.05), lending support to the causality from work-to-family conflict to intention to perform UPFB.

## 4. Discussion

In conclusion, Study 1 articulated that when employees experience high levels of work-to-family conflict, the intention of perpetrate UPFB would be increased. However, Study 1 has two limitations. First, it only tested the relationship between work-to-family conflict and UPFB, the underlying mechanism is under-examined. Second, given that nature of experimental methodology, the generalizability of our findings is limited. In an effort to address such shortcoming, a three-wave time-lagged filed study was conducted in Study 2. 

## 5. Study 2 Field Study

### Sample and Procedure

We collected our research data from 10 information technology companies in eastern provinces of China. We first contacted with human resource managers in the 10 information technology companies about the purposes of our research and got their permissions. We promised to share our research findings to exchange for non-interference data collection. Our research team went to the 10 companies and conducted the surveys. We announced an informed consent, which stated that our surveys were conducted with academic purpose and participants can engage in it voluntarily and the responses would be kept seriously confidential, to all participants before formal data collection. After this procedure, we distributed the paper questionnaire within an envelop with an unique code to each participant. Once completed, they were instructed to put the questionnaires into the envelop and then return it to our research team. 

To minimize potential common method variance (CMV), we separately collected research data in three waves [47]. At Time 1, we administered questionnaires to 380 participants, and employees rated work-to-family conflict and control variables (i.e., gender, age, education, organizational tenure, department, marital status, number of children living at home and annual household income). 345 employees’ research data were collected at Time 1, with a response rate of 90.79%. One month later, at Time 2, participants again reported their levels of guilt proneness, family motivation, and ethical leadership. 292 employees’ research data were collected at Time 2, with a response rate of 84.63%. At Time 3, one month after the Time 2 survey as well, we asked the participants to rate their levels of UPFB. 255 employees’ research data were collected at Time 3, with a response rate of 87.33%. Collectively, the total response rate of the data collection was 67.11%. 

In regard of demographic information, the average age of the participants was 27.84 years (SD = 7.77). Male accounted for 52.55%. Most of the participant held a master degree (46.67%). More detailed descriptive information was provided in Table 2.

## 6. Measures

We employed the translation-back translation procedure underscored by Brislin [45] to ensure the translated Chinese scales were semantically equivalent with the original scales. Unless otherwise mentioned, participants rated all the items with 5-point Likert scales (1 = “strongly disagree” to 5 = “strongly agree”). The alphas can be found in Table 2.

***Work-to-family conflict*.** Work-to-family conflict was measured using Netemeyer et al.’ [9] 5-item scale. Sample item includes “The demands of my work interfere with my home and family life.”

***Guilt proneness.*** As suggested by Chen and Moosmayer [48], we assessed the extent of employees’ guilt proneness by adopting 5 items selected from Cohen et al.’s [39] 16-item Guilty and Shame Proneness Scale (GASP). Sample item includes “You secretly commit a felony. What is the likelihood that you would feel remorse about breaking the law?” 

***Family motivation*.** Family motivation was measured using the 5-item scale developed by Menges et al. [18]. Sample item includes “I care about supporting my family.”

***Ethical leadership*.** We measured ethical leadership by adopting the 10-item scale developed by Brown et al. [17]. Sample item includes “The leader listens to what employees have to say.”

***UPFB*.** UPFB was accessed by the 7-item scale developed by Liu et al. [5] with a 5-point Likert scale (1 = “never” to 5 = “all the time”). Sample item includes “To help my family, I took company assets/supplies home for family use.”

***Control variables*.** Control variables used in Study 2 were consistent with Study 1.

## 7. Results

### 7.1. Confirmatory Factor Analyses

Since our research data were collected from the same source, we conducted confirmatory factor analyses (CFAs) to examine the distinctiveness of our major constructs by employing Mplus 8.0 [49]. The results were reported in Table 3, which showed that the five-factor measurement model (i.e., work-to-family conflict, guilt proneness, family motivation, UPFB, ethical leadership) exhibited a great model fit, with χ^2^ = 718.25, d*f* = 454, RMSEA = 0.05, SRMR = 0.05, CFI = 0.94, TLI = 0.94. Besides, we also tested a serial of competitive measurement models to compare with the baseline model (i.e., five-factor model). As presented, five-factor model showed superior model fit than all the other competitive measurement models. In conclusion, the discriminant validity of major variables was confirmed.

### 7.2. Hypotheses Testing

We conducted path analyses to examine our hypotheses by employing Mplus 8.0 [49], and results were shown in Figure 2 and Figure 3. Below we report the results from the analyses with control variables. Hypothesis 1 proposed that work-to-family conflict positively relates to UPFB. As shown, work-to-family conflict yielded a positive effect on UPFB (*direct effect* = 0.42, *SE* = 0.06, *p* < 0.01, R^2^ = 0.21), confirming Hypothesis 1.

Hypothesis 2 proposed that family motivation mediates the impact of work-to-family conflict on UPFB. Path analyses revealed that work-to-family conflict was positively related to family motivation (*β* = 0.46, *SE* = 0.06, *p* < 0.01; R^2^ = 0.21). Family motivation was also positively related to UPFB (*β* = 0.40, *SE* = 0.05, *p* < 0.01; R^2^ = 0.34). Furthermore, as suggested by Hayes [50], Bootstrapping technique was performed to test the mediating hypothesis. It is acknowledged that if the 95% confidential interval does not contain zero, researchers can conclude the indirect effect of work-to-family conflict on UPFB via family motivation was established [51]. Results showed that the indirect effect of work-to-family conflict on UPFB via family motivation was significantly positive (*indirect effect* = 0.18; *SE* = 0.03; 95% CI = [0.124, 0.257]), validating Hypothesis 2.

Hypothesis 3 predicted guilt proneness exerts a moderating effect on the relationship between work-to-family conflict and family motivation. The interaction term was constructed as the product of grand-mean centered work-to-family conflict and guilt proneness. The results revealed that the interaction between work-to-family conflict and guilt proneness was significantly and positively related family motivation (*β* = 0.11, *SE* = 0.05, *p* < 0.05; R^2^ = 0.30), suggesting that guilt proneness had a positive moderating effect on the above mentioned relationship. Specifically, as shown in Figure 4, the positive relationship between work-to-family conflict and family motivation was stronger for high guilt proneness (i.e., a standard deviation above the mean; *β* = 0.49, *SE* = 0.08) than low guilt proneness (a standard deviation below the mean; *β* = 0.29, *SE* = 0.06). Hence, Hypothesis 3 received empirical confirmation.

Hypothesis 4 proposed that guilt proneness intensifies the indirect relationship of work-to-family conflict on UPFB via family motivation. The indirect effect of work-to-family conflict on UPFB via family motivation was 0.14 with 95% CI = [0.080, 0.231] for high guilt proneness, and 0.09 with 95% CI = [0.044, 0.142] for low guilt proneness. The discrepancy of the indirect effects between the two conditions was 0.06, with 95%CI = [−0.004, 0.130]. Thus, Hypothesis 4 was confirmed.

Hypothesis 5 predicted ethical leadership mitigates the relationship between family motivation and UPFB. The interaction term was calculated as the product of grand-mean centered family motivation and ethical leadership. The results revealed that the interaction between family motivation and ethical leadership was significantly and negatively related UPFB (*β* = −0.11, *SE* = 0.04, *p* < 0.01; R^2^ = 0.32), implying that ethical leadership had a negative moderating effect on the relationship of those two variables. Specifically, as shown in Figure 5, the positive relationship between family motivation and UPFB was weaker when ethical leadership was high (*β* = 0.30, *SE* = 0.06) than when it was low (*β* = 0.49, *SE* = 0.07). Therefore, Hypothesis 5 was validated. 

Hypothesis 6 proposed that ethical leadership dampens the indirect relationship of work-to-family conflict on UPFB via family motivation. The indirect effect of work-to-family conflict on UPFB via family motivation was 0.14 with 95% CI = [0.077, 0.290] for high ethical leadership, and 0.22 with 95% CI = [0.147, 0.309] for low ethical leadership. The discrepancy of the indirect effects between the two conditions was −0.09, with 95% CI = [−0.145, −0.028]. Thus, Hypothesis 6 was supported. All results of bootstrapping analyses were provided in Table 4. Summary of hypotheses testing was presented in Table 5.

## 8. Discussion

The results of the field study lent further empirical support for our full theoretical model. In addition to replicating the findings that work-to-family conflict contributed to UPFB in the actual work settings, we also unveil that the relationship between work-to-family conflict and UPFB was mediated by family motivation. Besides, the relationship between work-to-family conflict and family motivation and the indirect relationship between work-to-family conflict and UPFB through family motivation were higher when guilt proneness was high. Moreover, the relationship between family motivation and UPFB and the indirect relationship from work-to-family conflict to UPFB via family motivation were weakened by ethical leadership. This study complements Study 1 to gain increased validity and generalizability.

## 9. General Discussion

Despite the prevalence of UPFB in the organizations, research concerning its antecedents is still in its infancy. The current study applied self-determination theory to shed light on how work-to-family leads to UPFB. Specifically, through our scenario-based experiment and field study, we confirmed that work-to-family conflict will drive employees to pursue family-related external aspirations and thus energize their family motivation [18]. Furthermore, strong family motivation predisposed employees to engaged in more UPFB. Additionally, our results also revealed that the effect of work-to-family conflict on family motivation hinged on employees’ guilt proneness. When guilt proneness was high, the effects of work-to-family conflict on family motivation and eventually on UPFB will be amplified. Moreover, we also identified that ethical leadership can dampen the positive effect resulted by family motivation and thus alleviate the detrimental effect. These findings offer crucial implications to both theory and practice.

## 10. Theoretical Implications

This paper devotes to make contributions to the extant literature in several aspects. First, our research also contributes to the antecedents of UPFB. Although prior research has emphasized that unethical behaviors will emerge in employees who are experiencing role conflict [7], little is known about the relationship between work-to-family conflict and unethical behavior, especially UPFB. Our research draws on self-determination theory to uncover that work-to-family conflict may act as a potent predictor that triggers employees’ motivational process and eventually contribute to UPFB. Such findings highlight that although modern organizations seek to explore employees’ potential by encouraging them to devote more time into work [52], involving work-to-family conflict may tax the organizations by predisposing employees to engage in UPFB and thus harm the interests of the organizations [5]. By doing so, our research not only narrows the theoretical gap between work-to-family conflict and UPFB, but expands the knowledge about detrimental influences resulted by work-to-family conflict in the workplace.

Additionally, extant literature about prosocial unethical behavior mainly draws attention to unethical pro-organizational behavior, whereas UPFB has not been examined until recently [3,5]. Moreover, empirical evidence has only confirmed that family financial pressure (a pure family issue) and workplace bullying (a pure work issue) as the antecedent of UPFB [3,5], other predictors at the interface between work and family remain unexplored. In an effort to narrow this theoretical gap, Liu et al. [5] has suggested scholars to debunk more antecedents of UPFB to understand its nomological network. Through our empirical studies, we advance the literature by providing work-to-family conflict could be a distal antecedent and family motivation could be a proximal antecedent of UPFB, broadening the theoretical scope about the predictors of UFPB to the literature. In particular, prior research regarding the antecedents of UPFB concentrated on either family [5] or organizational factors [3], our research extend the literature by elucidating that UPFB may be caused by the interface of work and family factors (i.e., work-to-family conflict). By doing so, our study thus enlarges the academic scope of why employees exhibit UPFB. 

Second, our research also contributes to the literature of work-to-family conflict. In addition to resource/stress/strain perspectives, few scholars have adopted other theoretical perspectives to investigate the consequences of work-to family conflict [53]. Further, most of the extant research concerning the negative outcomes of work-to-family conflict mainly concentrates on how it provokes detrimental psychological states or how it results in decreased constructive behaviors [54,55]. Our study adopts a novel lens—self-determination, to investigate how work-to-family conflict lead to UPFB, thus broadening the horizons of how to construe work-to-family conflict in the modern society.

Third, we echo past initiatives to advancing the theoretical development and empirical findings regarding family motivation [18]. Due to the novelty of this construct, extant research concerns about family motivation is still severely limited [12,18]. Evidence revealed that family motivation is energized by family financial strain [12], other motivators are left unexplored. Our research extends the literature by testing work-to-family conflict as an antecedent of family motivation and clarifies the explicit underlying mechanism using the perspective of self-determination. Furthermore, our research also demonstrates that UPFB may potentially led by family motivation because strong desire to benefit one’s family, echoing previous scholars’ suggestions to offer potential downsides of family motivation [12]. In the regard, contrary to prior research, which mainly concentrates on the desirable influences [18,56], our findings also confirm the unintended “dark side” effects of such pro-social motivation (i.e., family motivation) [12]. By doing so, our research offers a richer picture of family motivation literature. 

Fourth, by revealing how contextual factors and individual differences jointly shape enhanced family motivation, we advance the development of self-determination theory. Self-determination theory posits that the types of motivation experienced by employees can be impacted by their aspirations [14]. We contend that guilt proneness enhances employees’ motivation to achieve the extrinsic aspiration resulted by work-to-family conflict. Most of existing empirical studies concerning self-determination theory mainly concentrates on how autonomous motivation will be boosted though satisfying three types of basic needs [20]. Severely less research considered the activation of controlled motivation [12]. This is problematic because controlled motivation also plays a pivotal role in organizational functioning whereas it may potentially lead to undesirable outcomes simultaneously [14,18]. Our research systematically analyses a specific form of controlled motivation, namely family motivation [12]. We unveil how family motivation will be provoked and when this relationship will be stronger. By doing so, our research contributes to self-determination theory by shedding light on the predictors and boundary conditions regarding family motivation.

Moreover, we also contribute to a burgeoning body of research that investigates the short-term dark sides of guilt-related constructs. In essence, state guilt pertains to a negative emotion that may consume individual’s cognitive resources thereby inducing detrimental influences [57]. Recently, challenging the consensus that guilt also leads to morally laudable outcomes, experimental research has also revealed that guilt activated a short-term mindset that focuses on repairing interpersonal damage which may result in lying [19]. Our research echoes the nascent research that examines the dark sides of guilt-related constructs by revealing that guilt proneness may indirectly contribute to more prosocial unethical behavior (i.e., UPFB), adding to the literature about its potential downside effect.

Lastly, we contribute to the ethical leadership literature by emphasizing its moderating role in the relationships between family motivation and subsequent behavior. Previous work entails ethical leadership mainly probed how such leadership generates the beneficial effects on positive job attitudes and morally praiseworthy behaviors [40,58], comparatively less research has examined ethical leadership as an effective moderating condition [43]. Our findings demonstrate that ethical leadership has a critical role to play beyond immediately contributing to performance [40]. Specifically, our research reveals that ethical leadership can be a boundary condition which will mitigate the influence of family motivation on UPFB, confirming that ethical leadership can be a desirable leadership in curtailing employees’ unethical behaviors in the organizations [17]. Our research highlighted the effect of employees’ motivation in driving unethical behavior can be varied depending on ethical contextual differences [13]. Our research also contributes to the business ethics literature, which notes that “dearth of research on interventions” that can discipline unethical behavior (p. 273) [59]. 

## 11. Practical Implications

First, in addition to exerting harmful influences on employees [9], our research reveals that work-to-family conflict may also immediately diminish the interests of organizations by impelling employees who involvomh work-to-family conflict to perpetrate UPFB. Prior research highlighted organizations can benefit from helping employees balance their work and family [8]. Therefore, we urge employers to minimize the extent of work-to-family conflict in the organizations, such as providing flexible arrangements with regard to how and where work is completed [60], carry out family-responsive policies [61]. By doing so, organizations can mitigate the negative effects of work-to-family conflict to some extent.

Second, by shedding light on the underlying mechanism of how work-to-family conflict led to UPFB, we notice that families have far-reaching meaningfulness for employees, namely, engaging in unethical behavior for the sake of families at all cost [5]. Prior research also highlighted families also serve as a potent sources for employees to increase work efforts [12]. Thus, organizations should pay increased attention to the significance of families for employees. For instance, organization can provide family-friendly policies or other measures to offer more support and care to employees’ families, thereby enhancing employees the senses of belonging, and minimizing the probability of benefiting families from deteriorating organizational interests.

Third, our research highlights the importance of hiring or selecting managers or team leaders behaving in ethical leadership style. On the one hand, in our research, we have revealed that ethical leadership, which can influence employees’ behavioral choosing process through ethical role modelling and atmosphere forming, dampening the relationship of family motivation and morally inappropriate behavior. Moreover, substantial studies have examined ethical leadership as an antecedent that leverage its beneficial effect on employees’ work outcomes, such as organizational citizenship behavior [62], ethical behaviors and job performance [40]. Therefore, hiring or selecting managers high in ethical leadership can benefit the organizations in both reducing unethical behaviors and facilitating instrumental outcomes, especially when work-to-family conflict is inevitable in some positions. Besides, drawing on mechanisms of how ethical leadership influences employees’ behaviors choosing process (ethical role modelling and atmosphere forming), top level managers should integrate ethics into organizational culture or set explicit moral standards to emphasize the importance of moral behavior [63].

### Existing Limitations and Future Directions

This paper has several limitations that require further addressed. Firstly, since data source were single in our filed survey, there may be potential concern about CMV [47]. Although the Harmon single-factor test indicated there were not serious CMV in our research, we still suggest future research to optimize the research design by employing multi-source data collection to lessen the concern of CMV, which is beneficial for improve the validity of the research findings.

Secondly, in regard of our scenario experiment, we did not examine the causality of work-to-family conflict and psychological responses due to concern of CMV (if measuring psychological responses and UPFB simultaneously) [47]. In our scenario experiment, we mainly focused on the causality of work-to-family conflict and UPFB to examine the core relationship of our theoretical model, future research can benefit by examining the causality of work-to-family conflict and its psychological responses and the causality of psychological responses and UPFB.

Thirdly, both two empirical studies were conducted in the Chinese culture context, which constrains the generalizability of our findings. As the long-held understanding, Chinese society and Chinese traditional culture strongly emphasizes individuals’ family obligations [12], making family motivation to exhibit pronounced behavioral effects. The notion of work and family lives construed in the norms of Confucian philosophy may be distinct from Western Cultures [60]. Future research can be conducted in different cultural contexts to generalize the research findings.

Fourthly, with respect to the relationship between work-to-family conflict and UPFB, we only considered the mechanism by employing the perspective of self-determination. We still think that we do not capture the complete picture of the underlying mechanisms of work-to-family conflict and UPFB. For instance, from an exchange perspective, work-to-family conflict threats the reciprocity of employees and their family members in some respects. Additionally, future research can also theorize and empirically examine the indirect effect of work-to-family conflict on UPFB in other theoretical perspectives or employing other mediators to provide more nuanced insight into the underlying mechanisms. 

Lastly, we suggest that it is worthwhile for investigating more boundary conditions in the association between work-to-family conflict and UPFB. For instance, we only focus on the individual difference that can intensify the relationship between work-to-family conflict and family motivation and eventually UPFB, it is also important to examine the boundary conditions that can dampen the effect of work-to-family conflict on family motivation and eventually decrease UPFB, such as organizational identification [4]. When considering the boundary conditions that influence the relationship between family motivation and UPFB, we only investigated ethical leadership as a mitigating role in attenuating the relationship the two and the indirect relationship between work-to-family conflict and UPFB via family motivation, there may be some other potential exacerbating factors that may amplify the detrimental effect of work-to-family conflict and UPFB, such as employees’ family centrality (captures as the extent of importance ascribed to family roles for employees) [64]. It is also essential to uncover such factors so as to weaken their interference.

## 12. Conclusions

By adopting the lens of self-determination, we examined whether, why, and when work-to-family conflict increases employees’ UPFB. Our empirical results revealed that employees’ family motivation would be enhanced when they confronted work-to-family conflict, especially when they possessed strong guilt proneness. Family motivation, in turn, motivated employees to commit UPFB to benefit their family members, especially when ethical leadership is low in the workplace. Going beyond the existing literature, which investigates the antecedents of UPFB by focusing on either work or family domains separately, our research sheds light on the interface of work and family (i.e., work-to-family conflict) that contributes to the emergence of UPFB through a motivational pathway. Additionally, we also advance a nascent vein of research examining the potential downsides of family motivation and guilt proneness at work. Nevertheless, we recognize that we have only taken a small step in exploring these relevant issues. We encourage further research to extend scholarly insights concerning these topics.

## Figures and Tables

**Figure 1 ijerph-20-04368-f001:**
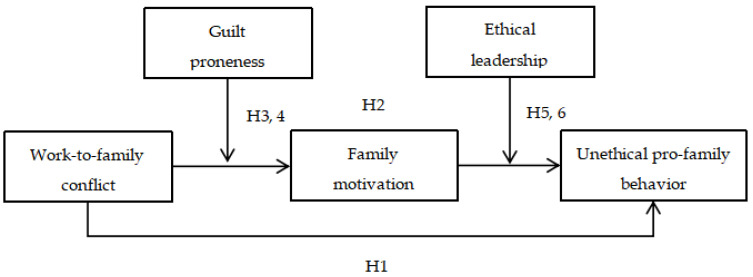
Theoretical Model.

**Figure 2 ijerph-20-04368-f002:**
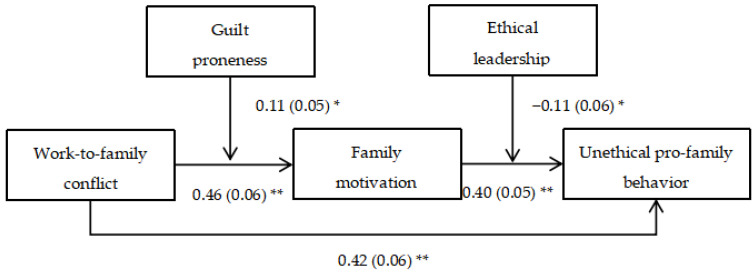
Unstandardized Estimates of Path Analyses. Note: *N* = 255. For brevity, we did not present the coefficients of the control variables (i.e., gender, age, education, tenure, marital status, and the number of children). * *p* < 0.05, ** *p* < 0.01, two-tailed test.

**Figure 3 ijerph-20-04368-f003:**
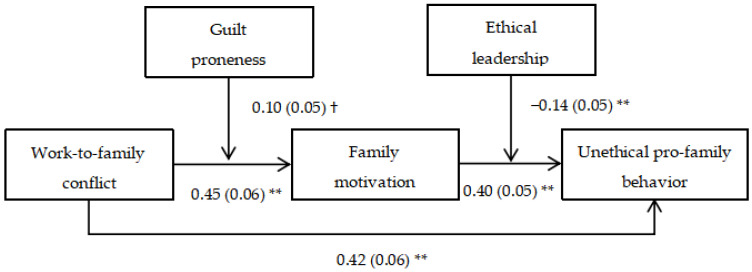
Unstandardized Estimates of Path Analyses without Control Variables. Note: *N* = 255. † = 0.0056.** *p* < 0.01, two-tailed test.

**Figure 4 ijerph-20-04368-f004:**
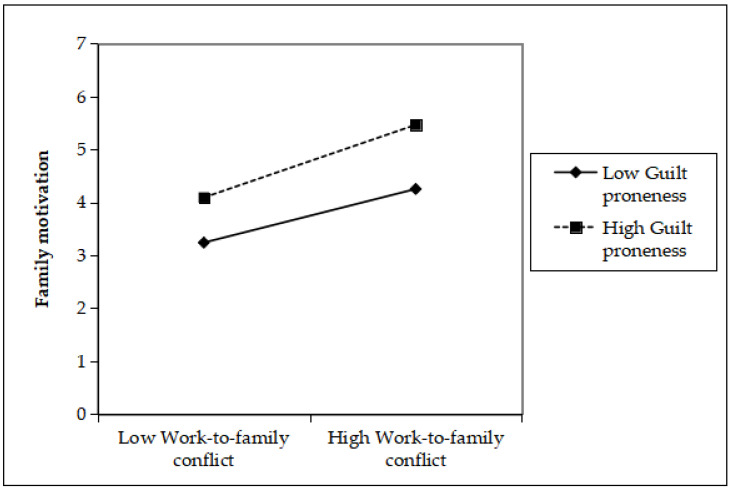
The Moderating Effect of Guilt Proneness.

**Figure 5 ijerph-20-04368-f005:**
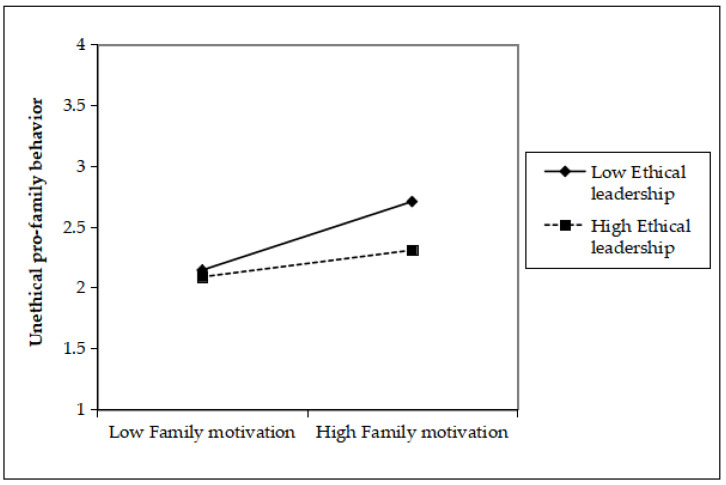
The Moderating Effect of Ethical Leadership.

**Table 1 ijerph-20-04368-t001:** Descriptive Statistics in Study 1.

Variable	1	2	3	4	5	6	7	8
1. Age								
2. Gender	0.13							
3. Education	0.03	−0.10						
4. Tenure	0.53 **	0.17	−0.22 *					
5. Marital status	0.32 **	0.09	0.04	0.17				
6. Children	0.22 *	0.15	−0.13	0.33 **	0.58 **			
7. WFC	−0.09	−0.05	−0.01	−0.11	−0.05	−0.01		
8. UPFB intention	0.05	−0.07	0.18	−0.15	−0.10	−0.14	0.24 *	(0.86)
M	30.96	0.47	2.14	3.80	2.74	0.92	0.49	1.83
SD	4.52	0.50	0.51	0.65	0.67	0.60	0.50	0.66

Note: *N* = 118. WFC: work-to-family conflict; UPFB: intention to perform unethical pro-family behavior; Children: the number of children at home. Alphas were presented in the diagonal. * *p* < 0.05, ** *p* < 0.01, two-tailed test.

**Table 2 ijerph-20-04368-t002:** Descriptive Statistics in the Study 2.

Variable	1	2	3	4	5	6	7	8	9	10	11
1.Age											
2.Gender	0.13 *										
3.Education	−0.03	0.07									
4.Tenure	−0.04	−0.01	−0.02								
5.Marital status	0.22 *	0.10	−0.04	−0.10							
6.Children	0.42 **	0.08	−0.12	−0.13 *	0.66 **						
7.WFC	−0.04	0.03	−0.12	0.05	0.01	0.08	(0.87)				
8.GP	0.02	−0.04	−0.15 *	0.00	0.00	0.05	0.30 **	(0.83)			
9.FM	0.00	0.00	−0.04	0.07	0.05	0.02	0.45 **	0.40 **	(0.89)		
10.EL	−0.02	0.08	0.08	−0.12	0.01	0.00	0.27 **	0.37 **	0.29 **	(0.86)	
11.UPFB	0.01	−0.11	−0.11	0.04	0.06	0.09	0.43 **	0.52 **	0.52 **	0.23 **	(0.90)
M	1.78	0.53	2.77	2.89	2.13	0.33	2.49	2.42	2.49	2.43	2.55
SD	0.77	0.50	0.96	0.92	0.92	0.60	0.91	0.90	0.92	0.85	0.89

Note: *N* = 255. WFC: work-to-family conflict; GP: guilt proneness; FM: family motivation; EL: ethical leadership; UPFB: unethical pro-family behavior; Children: the number of children at home. Alphas were presented in the diagonal. * *p* < 0.05, ** *p* < 0.01, two-tailed test.

**Table 3 ijerph-20-04368-t003:** Confirmatory Factor Analyses Results.

Model	χ^2^	D*f*	Δχ^2^	RMSEA	SRMR	CFI	TFI
Five-factor	721.70	454		0.05	0.05	0.94	0.94
Four-factor	1092.68	458	370.98 ***	0.07	0.07	0.87	0.86
Three-factor	1440.70	461	719.00 ***	0.09	0.08	0.79	0.78
Two-factor	1730.35	463	1008.65 ***	0.10	0.09	0.73	0.71
One-factor	2848.71	464	2127.01 ***	0.14	0.14	0.50	0.46

Note: *N* = 255. Four-factor model: combined work-to-family conflict (WFC) and guilt proneness (GP). Three-factor: combined WFC, GP, and family motivation (FM). Two factor model: combined WFC, GP, FM, and ethical leadership. One factor model: combined all the five variables. *** *p* < 0.001, two-tailed test.

**Table 4 ijerph-20-04368-t004:** Bootstrapping Moderated Mediation Effect Test Results.

Path	Moderator	Estimate	95% Confidential Interval	
			Lower	Higher
Work-to-family conflict→family motivation→unethical pro-family behavior	Lower GP (M − SD)	0.09	0.044	0.142
	Higher GP (M + SD)	0.14	0.080	0.231
	Difference	0.06	0.004	0.130
	Lower EL (M − SD)	0.22	0.147	0.309
	Higher EL (M + SD)	0.14	0.077	0.209
	Difference	−0.09	−0.145	−0.028

Note: *N* = 255. GP: guilt proneness; EL: ethical leadership. Bootstrapping randomly sampled 5000 times. Estimates were unstandardized.

**Table 5 ijerph-20-04368-t005:** Summery of Hypotheses Testing.

Hypothesis	Supported	
	Experiment	Survey
H1: Work-to-family conflict is positively related to UPFB	Yes	Yes
H2: Family motivation mediates the relationship between work-to-family conflict and UPFB.	-	Yes
H3: Guilt proneness moderates the relationship between work-to-family conflict and family motivation, such that the relationship is stronger when guilt proneness is high rather than when is low.	-	Yes
H4: Guilt proneness moderates the indirectly relationship between work-to-family conflict and UPFB via family motivation, such that the indirect relationship is stronger when guilt proneness is high rather than when is low.	-	Yes
H5: Ethical leadership moderates the positive relationship between family motivation and UPFB, such relationship will be weaker when ethical leadership is high rather than when it is low.	-	Yes
H6: Ethical leadership moderates the indirect relationship between work-to-family conflict and UPFB via family motivation, such that the indirect relationship will be weaker when ethical leadership is high rather than when it is low.	-	Yes

## Data Availability

Due to the confidentiality of the respondents, the data of are not publicly available.

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
