# Peer review of "My Family Accounts Much for Me: How Does Work-to-Family Conflict Lead to Unethical Pro-Family Behavior"

_ijerph, 2023, doi:10.3390/ijerph20054368_

Round 1

Reviewer 1 Report

Minor writing issues which can be corrected with proofreading. Examples include:

p1-lines 26-27 "Over the past decades, numerous corporate scandals and unethical incidents in the organizations emerge in an endless stream"  (should be past-tense)

Analysis issue:

table 1 shows gender, a categorical variable, used in correlation analysis. While a dummy variable for gender can be used in regression as a yes/no variable it should not be used in regression because regression needs continuous variables. Ordinal variables can be used in correlation if ordinal categories are equal.

p8-Line 386-387, it will help the reader if you describe how you determined the two groups such as above/below the median, above/below the mean, etc. 

Reviewer 2 Report

Please address the following issues,

Mention the software which you have used. 2. Display the picture of path analysis rather than a table. 3. Make a separate table from where one can see the acception or rejection of the hypothesis. 4. Mention the direct and indirect effects of the variables. 5. There is 30% plagiarism in your paper, please reduce it to less than 20%. Plagiarism report has been attached for your reference

Reviewer 3 Report

1. Figure 1. The theoretical model does not seem to be displayed perfectly, it is recommended to correct the format.

2. It is recommended that the research model be marked with Hypothesis 1-6, so that readers can quickly understand the process of the entire research implementation.

3. It is suggested that the content of the Informed Consent Statement for participation in Study 1 Scenario Experiment and Study 2 Field Study be added to the article.

4. It is recommended to detail the meanings represented by Table 1 and Table 2 for Means, Standard Deviations, and Correlations of Variables.

5. Is there a pre-test of reliability and validity for the items of the questionnaire?

6. Some of the topics of this research are controversial, and the participants may not express the real situation. It is suggested to explain how to obtain more real answers from the participants.

Round 2

Reviewer 2 Report

I think authors have addressed all the points successfully so I recommend to accept the paper after making following small changes.

1. Improve the conclusion part, it should be synchronised with the results.

2. Compare your results with existing literature.
